

# Webcam network and image database for studies of phenological changes of vegetation and snow cover in Finland, image time series from 2014-2016

5  Mikko Peltoniemi[1],*, Mika Aurela[2], Kristin Böttcher[3], Pasi Kolari[4], John Loehr[5], Jouni Karhu[6], Maiju Linkosalmi[2], Cemal Melih Tanis[2], Juha-Pekka Tuovinen[2], Ali Nadir Arslan[2]

1    Natural Resources Institute Finland (Luke), Latokartanonkaari 9, FIN-00790, Helsinki, Finland, mikko.peltoniemi@luke.fi

10  2    Finnish Meteorological Institute, Erik Palménin aukio 1, FI-00560, ali.nadir.arslan@fmi.fi, mika.aurela@fmi.fi

3    Finnish Environment Institute (SYKE), Mechelininkatu 34a, FIN-00251 Helsinki, Finland, kristin.bottcher@ymparisto.fi

Department of Physics, PO Box 68, 00014 University of Helsinki, pasi.kolari@helsinki.fi

Lammi Biological Station, University of Helsinki, Pääjärventie 320, 16900 Lammi, Finland, john.loehr@helsinki.fi

6    Natural Resources Institute Finland (Luke), Paavo Havaksen tie 3, 90014 Oulun yliopisto, Jouni.karhu@luke.fi

Correspondence to: Mikko Peltoniemi (mikko.peltoniemi@luke.fi), Ali Nadir Arslan (ali.nadir.arslan@fmi.fi)

**Abstract.** In recent years, monitoring of the status of ecosystems using low-cost web (IP) or time lapse cameras has received wide interest. Networked cameras can provide information about snow cover and vegetation status with a broad spatial coverage and high temporal resolution, and serve as ground truths to earth observations, and be useful for gap-filling of

cloudy areas in earth observation time series. Networked cameras can also play an important role in supplementing laborious phenological field surveys and citizen-science projects, which also suffer from observer-dependent observation bias. We established a network of digital surveillance cameras for automated monitoring of phenological activity of vegetation and snow cover in the boreal ecosystems of Finland. Cameras were mounted at 14 sites, each site having 1-3 cameras. Here, we document the network, basic camera information and access to images (see, https://doi.org/10.5281/zenodo.777952) in the

permanent data repository (www.zenodo.org/communities/phenology_camera/). Individual DOI-referenced image time series from cameras are consisted of half-hourly images collected between 2014 and 2016. Additionally, we present example colour index time series derived from image time series from two contrasting sites.

## 1 Introduction

Year-to-year variation in the phenological cycle of seasonal ecosystems is large. The snow cover and the timing of snowmelt

vary considerable from year to year, vegetation follows the temporal variation of temperatures, and budburst dates can vary



by 40 days (Häkkinen, 1999), but our ability to predict this variation is limited (Basler et al., 2016). Consequently, phenology remains among the key components causing uncertainties in our estimates of vegetation carbon balances (Richardson et al., 2013). Obviously, more monitoring data is required for improving the understanding of the sources of phenological variation and to directly support its monitoring, as present methods for field and snow monitoring of phenology

are laborious and costly. Consumer-grade cameras provide an interesting opportunity to monitor the seasonal cycle of ecosystems because they record structural information at high spatial resolution, and in doing so, time series of images can provide useful information about temporal changes in ecosystems with low costs.

In recent years, the ecological community has started to harvest image time series for the purposes of phenological analysis
(Richardson et al., 2007). In these analyses, typically some sub-region of the image is followed for color changes in time, using indices representing phenological changes of leaf colour. Analyses of colour changes and phenological transition dates can be easily automated by using statistical methods that identify break points in time series (Elmore et al., 2014; Klosterman et al., 2014; Wingate et al., 2015). Although colours of the images in the time series are affected by illumination angle, cloud cover and camera type, much of the potential problems in phenological times series can be largely avoided by
using images from similar environmental conditions, and by selecting and colour indexing appropriately. For example average chromatic coordinates for a sub-region of the image have been aggregated/averaged over images taken over one to three days by concentrating on well exposed midday images (e.g. Sonnentag et al., 2012; Linkosalmi et al., 2016). Green chromatic coordinate, GCC = G / (R+B+G), on the other hand, is a fairly robust index of colours, because the opposite changes in red and blue balance partially balance each other over the course of the day (Wingate et al, 2015).

Cameras have been deployed on a range of ecosystems, including deciduous and mixed species ecosystems (Richardson et al., 2007, Ahrends et al., 2009, Sonnentag et al., 2012, Mizunuma et al., 2013), grasslands (e.g. Migliavacca et al., 2011), peatlands (Westergaard-Nielsen et al., 2013, Peichl et al., 2015, Linkosalmi et al., 2016) and coniferous forests (Nagai et al., 2012, Linkosalmi et al., 2016). Deciduous budburst and leaf senescence events and their relationship with CO2 exchange
have been a focus in a number of studies (Richardson et al., 2007, Ahrends et al., 2009, Sonnentag et al., 2012, Mizunuma et al., 2013, Wingate et al., 2015). Reasons for plant colour changes have been related to the increase of green biomass, and also to the development of biochemical development of leaves occurring over the season (Keenan et al., 2014; Yang et al., 2014). While more detailed radiative transfer analyses may be required to confirm the consequences of pigment and other changing properties of leaves on the colours (Wingate et al., 2015), the most clear phenomena such as budburst and autumn
colour peaks can already be extracted with simple time series and break points analyses with fairly good accuracy (Peltoniemi et al., submitted). Besides vegetation monitoring, time lapse images have been used in snow cover monitoring. Salvatori et al. (2011) developed a method for analyzing snow cover changes based on blue channel histogram. The method has been recently tested also in northern boreal Finland (Arslan et al., submitted). Perhaps the most obvious use of camera time series stems from the possibility of using time lapse material for quality assurance and documenting the site history. As



such, they provide supportive information on the ambient conditions, i.e. information about weather, vegetation and unexpected disturbances.

Given the wealth of features that can be extracted from image time series of cameras, and the potential of introducing
cameras to cover wide areas with high density, it is not surprising there is a large interest in using them as ground references for earth observation (EO) products. A particular benefit of cameras is that they can provide continuous time series of reference observations, as they are not or are rarely occluded by clouds. Previous research has already shown good correspondence between the camera-based and EO phenology products (Zhang et al., 2006, Morisette et al., 2009, Elmore et al., 2012, Hufkens at al., 2012, Klosterman et al., 2014).

The full benefits of using cameras in ecosystem monitoring are reached when information from multiple cameras are analyzed together, and thus many of the cameras have been set up in smaller or larger networks of sites. In the US, PhenoCam network has collected time lapse images for several years (Richardson et al. 2009; phenocam.sr.unh.edu). In Japan researchers have collected images from atmosphere–ecosystem carbon flux sites within the framework of Japan
Phenological Eyes Network (PEN; Nasahara and Nagai 2015; pen.agbi.tsukuba.ac.jp) with the aim of linking them to other remote sensing and other ecosystem data. The European network of carbon flux sites operate cameras at least in 50 sites (Wingate et al., 2015). Phenological camera monitoring is being more widely implemented within the framework of the US National Ecological Observatory Network (NEON) and the European Union's Integrated Carbon Observation System (ICOS). Many networks (E.g., Australian Phenocam Networkwww.phenocam.org.au) and ICOS are providing data online
by request and/or have plans for open distribution of images (Brown et al., 2016).

Keeping the benefits of using cameras for near-ecosystem remote sensing in mind, and in order to contribute to the growing body of cameras monitoring ecosystems of the earth, we established a network of cameras for monitoring boreal ecosystems in Finland. Here, we document the sites of this monitoring network including the equipment used, describe the availability of
data in open repositories, as well as report our first hand experiences while setting up the network and envisage further development needs of the network. We further made image data public from altogether 27 cameras from 14 sites in Zenodo data archive (https://www.zenodo.org/) established by EU (EU OpenAire and CERN) under Creative Commons Attribute 4.0 license, appended with metadata sheets fulfilling the criteria developed by Brown et al. (2016) for integrated camera networks.

**2 Copyright statement**

Material in the database is published under Creative Commons Attribute 4.0 license. For the availability of newer images that are not yet transferred to the open repository, and for ancillary data, contact authors, see section Data availability.



# 3 Network of cameras

## 3.1 Cameras and installations

One to three cameras were installed for the sites depending on the canopy conditions at the site (Table 1, see also
http://monimet.fmi.fi/?_page=Cameras). The cameras were mounted to provide views of the canopy, ground and/or a wider

landscape. Some cameras were mounted closer to individual tree crowns, thus narrowing the region of interest (ROI) of
image analysis to the target trees without confounding contribution from the ground. The cameras mounted below the
canopy monitor the phenology of understorey vegetation and snow cover. At the wetland sites, a single camera fulfills all
these purposes. Many of the cameras allow defining multiple ROIs to monitor various species and phenomena at the same
time.

All cameras are set to a fixed white balance and automatic exposure, targeted northwards where feasible depending on the
view and mounting options, and triggered for half-hourly submission of snapshots to a ftp server, excluding the night hours.
Images are taken at maximum or quarter of the maximum resolution. Image jpegs are uncompressed. Most of the sites and
analyses of this study used image time series acquired with StarDot NetCam SC5 cameras, while more recently also an

AXIS P1357E camera was installed at one the sites (Paljakka). We did not expect color reproduction of cameras to be
standardized and calibrated, i.e. comparable across cameras, so we allowed some variation in the camera settings by local
conditions.  In StarDot cameras R, G, and B channels amplification have been modified from default settings (but are kept as
fixed) by local conditions so that cameras produce visually approximately realistic colours in typical conditions. Pixel digital
numbers are not thus directly comparable across cameras, but allow camera specific analyses of e.g. temporal development

of digital numbers and indices based on them. Iris and zoom were fixed to camera specific values depending on the view of
the camera. Cameras adjust for brightness of the images automatically. In AXIS P1357E, similar settings were used, most
importantly; white balance was set to fixed ('Fixed outdoor 1' in the camera menu). Selected cameras also have grey
reference plates to monitor the stability of colour reproduction of cameras.

## 3.2 Sites

The locations of sites range from the hemiboreal (Tvärminne) to the northern boreal (Kaamanen) phytogeographical zone
(Figure 1, Table 1), with views varying by ecosystem type and the position of the camera in the site (Figure **2**). All sites
belong to the subarctic climate zone (Dfc) according to Köppen's climate classification, except the Tvärminne site which is
situated in the warm-summer humid continental climate zone (Dfb). The start and duration of continuous snow cover varies
along the climate gradient from south-west to north in Finland. Permanent snow cover arrives on average (normal period

1981-2010) in October and January and melts in May and March for the Kaamanen and Tvärminne sites, respectively. The
thermal growing season begins when mean daily temperatures exceed 5°C at the end of April in southern Finland and at the



end of May in northern Lapland. Temperatures decrease below 5°C ending the thermal growing season at the end of September in the north and in late October or beginning of November in southwestern Finland.

The cameras were installed at existing monitoring and research sites. These sites include ICOS and other eddy-covariance
sites, LTER (Long –Term Ecosystem Research) and ICP Forests Level II sites, and other sites with previous phenological monitoring. Some sites are highly instrumented and intensively monitored ecosystem research sites such as Hyytiälä SMEAR II, Värriö SMEAR I, and Sodankylä eddy-covariance sites, while others are less intensively monitored. For every site there is at least a meteorological station (except Punkaharju, see below), and varying number and types of ancillary monitoring measurements. Data from Finnish Meteorological Institutes (FMI) meteorological stations can be freely
downloaded from the FMI open data portal (https://en.ilmatieteenlaitos.fi/open-data), e.g. by using a user interface 'FMI weather data downloader v0.15' (Salmi, 2016).

### 3.2.1 Hyytiälä

Hyytiälä site is situated at the University of Helsinki's Hyytiälä Forestry Field Station in an even-aged Scots pine (*Pinus sylvestris*) stand with scattered spruce and deciduous trees. One camera is focused on the pine canopy with one birch crown
in view. The second camera monitors the forest floor which is dominated by dwarf shrubs and feather mosses. The camera observations are supported by a wide selection of continuous measurements of meteorology, gas exchange, tree ecophysiology and soil at SMEAR II research station (Hari and Kulmala 2005, http://www.atm.helsinki.fi/SMEAR). These data are available at AVAA portal (https://avaa.tdata.fi/web/smart/smear/).

### 3.2.2. Kaamanen

The Kaamanen wetland is an open mesotrophic fen within the aapa mire region in northern Finland. The surface pattern of these northern fens (aapa mires) consists typically of wet flarks and drier strings. The flarks are most of the time inundated, but the strings with a height of approximately 0.8 m are constantly above the water table covering about 40% of the fen surface. The site has no permafrost, but thin lenses of ice may remain in the well-insulated strings until late summer. The
flarks are covered by different sedges (*Carex* spp.) and moss species, while the higher strings are dominated by various shrubs, such as *Ledum palustre*, *Empetrum nigrum*, *Rubus chamaemorus* and *Betula nana*. The maximum single-sided LAI was estimated to be 0.7 (Aurela et al., 2001). The $CO_2$ and $CH_4$ fluxes between the fen and the atmosphere, as well as a variety of environmental parameters (air and soil meteorology together with different radiation components), are continuously measured at the site.


### 3.2.3 Kenttärova

The Kenttärova site is situated in a Norway spruce forest at Pallas, where it is operated in conjunction with the Pallas–Sodankylä Global Atmosphere Watch station. The dominant tree height of 14.5 and an LAI of 2.0 $m^2$ $m^{-2}$ were estimated for the spruce forest in 2011. The minor population of *Betula pubescens* has an LAI of 0.1 m2 m–2. The main species of the ground floor are *Vaccinium myrtillus, Empetrum nigrum, Vaccinium vitis-idaea* and the forest mosses *Pleurozium schreberi,*

*Hylocomium splendens*, and *Dicranum polysetum*. Carbon dioxide fluxes together with an extensive set of meteorological parameters are measured at the site. Kenttärova is an associated site within the ICOS flux network.

### 3.2.4 Lammi

The Lammi site is situated at University of Helsinki's Lammi Biological Station (LBS) which belongs to the Finnish Long-

Term Socio-Ecological Research network (FinLTSER). The Station is located in a region characterized by boreal forest, lake and agricultural landscapes. Three cameras are present (ground, crown and landscape) including deciduous species *Betula pubescens, Acer platanoides, Ulmus glabra, Prunus padus, Tilia cordata* and *Ribes alpinum.*. The FMI weather station 'Hämeenlinna Lammi Pappila' is located at LBS. LBS also maintains a Photosynthetically Active Radiation sensor (data available upon request from LBS).

### 3.2.5 Lompolojänkkä

The Lompolojänkkä site is located on an open, mesotrophic sedge fen. The field layer vegetation in the wetter parts of the fen is dominated by sedges, while dryer parts are characterized by fairly dense stands of *Betula nana*. Low shrubs can be found across the fen with relatively low coverages. The midsummer mean vegetation height on the fen is 40 cm, and the one-

sided LAI is 1.3 $m^2$ $m^{-2}$. For a more detailed description of Lompolojänkkä, see Aurela et al. (2009) and Lohila et al. (2010). The camera was installed so as to provide a general view on the dryer part of the fen. The CO2 and CH4 fluxes together with a set of environmental parameters (air and soil meteorology together with different radiation components) are continuously measured at the site. Kenttärova is a Class 2 site within the ICOS flux network.

### 3.2.6 Paljakka

Paljakka is phenology monitoring site in central Finland belonging to the phenological monitoring network operating in Finland (Poikolainen et al., 1996; Pudas et al., 2008), where seasonal development of main tree species in Finland is monitored. The camera is located at station roof, and is focused on a Norway spruce stand. Image view includes small birch trees that are monitored for phenology. The FMI weather station 'Puolanka Paljakka' is located in the immediate vicinity of

the camera.

### 3.2.7 Parkano

Paljakka is a phenology monitoring site in southern Finland, belonging to the same network of sites as Paljakka site. The camera is located at station roof, and is focused on a mixed forest landscape including trees that are followed for phenology, and a lake behind the trees. The FMI weather stations ('Karvia Alkkia') is within approx. 20 km distance.

### 3.2.8 Punkaharju

The site is a mature spruce (*Picea abies*) stand on mesic soil. It host three cameras at different heights, one taking images of the ground ground, one at the crown level, and one taking images of surrounding landscape.  The nearest FMI weather station 'Savonlinna Punkaharju Laukansaari' is within 3 km distance. The Punkaharju site belongs to the ICP Forests (International Cooperative Programme on the Assessment and Monitoring of Air Pollution Effects on Forests) level II

monitoring site for long term ecosystem monitoring. A diverse set of ecosystem monitoring is conducted at the site, see ICP Forests manual for details and monitoring information collected (http://icp-forests.net/page/icp-forests-manual).

### 3.2.9 Sodankylä forest

The Scots pine (*Pinus sylvestris*) forest site in Sodankylä is situated within the Arctic Research Centre of Finnish Meteorological Institute. The dominant tree height is 13 m and the forest has a total LAI of 3.6. The sparse ground vegetation consists of lichens (73%), mosses (12%) and ericaceous shrubs (15%). There are three cameras at this site, with views or the forest canopy, crown and the ground, respectively. Carbon dioxide fluxes together with an extensive set of meteorological parameters are measured at the site. The Sodankylä forest is a Class 1 site within the ICOS flux network.

### 3.2.10 Sodankylä wetland

The Sodankylä wetland (Halssiaapa) is an open fen in the vicinity of the Sodankylä forest flux site. It is dominated by large, treeless flarks with abundant sedge vegetation, complemented by fairly low and narrow ridges with birch trees (of 5-7 m in height). $CO_2$ and $CH_4$ fluxes as well as basic meteorological parameters are

measured at the site.

### 3.2.11 Suonenjoki

Suonenjoki site has a Scots pine (*P. sylvestris*) stand on sub-xeric soil. There is a weather stations maintained by Natural Resources Institute Finland at the site (data on request from the corresponding author). The nearest FMI weather station

'Suonenjoki Iisvesi' is within 3 km distance.  No other monitoring actions are conducted at this site.

### 3.2.12 Tammela

The site presently hosts two cameras, one taking images of the ground, and another taking images of the surrounding landscape. Tammela site has a mature spruce (*P. abies*) stand on mesic soil. It belongs to the same network of ICP Forests



level II sites. Consult ICP II Forests manual for permanent monitoring data collected (http://icp-forests.net/page/icp-forests-manual). The FMI weather station 'Somero Salkola' is in the site.

### 3.2.13 Tvärminne

The Tvärminne site belongs to the Finnish Long-Term Socio-Ecological Research network (FinLTSER) and is located at the western Gulf of Finland at the University of Helsinki's Tvärminne Zoological Station. The area was postglacially uplifted above the sea level and is characterized by a mix of bedrock and clay soil material and herb-rich forest (nemoral) forest. One camera is installed on the roof of the Station's main building viewing the landscape at the shoreline of the archipelago (Figure 2, af). The camera allows the monitoring of the tree phenology, the seasonal development of reeds and the evolution of sea ice. The tree types in the camera's field of view include elm (*Ulmus glabra*), maple (*A. platanoides*), downy birch (*B. pubescens*), ash (*Fraxinus excelsior*), common alder (*Alnus glutinosa*) and Scots pine (*P. sylvestris* L.). The weather station 'Hanko-Tvärminne' of the FMI is located at the site and provides measurements of air temperature, precipitation, humidity and snow depth.

### 3.2.14 Värriö

The Värriö site is located in Salla, eastern Lapland, at the northern alpine timberline on the summit plateau of a hill. The relatively open forest stand is dominated by Scots pine (*P. sylvestris*) of various age with occasional mountain birch (*Betula pubescens* var. *punilla*) in the understorey. The forest floor vegetation consists of a variety of mosses, lichens and dwarf shrubs. The site accommodates three cameras for taking images of the landscape, the pine crowns and the forest floor. The cameras are part of SMEAR I research station (Hari et al. 1994, http://www.atm.helsinki.fi/SMEAR) where continuous measurements of meteorology, gas exchange, tree ecophysiology and soil are performed. The data are available at AVAA portal (https://avaa.tdata.fi/web/smart/smear/).

## 4 Examples of processed image data

Here we show examples of the data collected within our network, focusing on the effect of systematic change in illumination conditions during the annual cycle of solar elevation known to influence colour spectra of incoming radiation, which potentially has effects on the calculated colour ratios from images. The analyses were done for five regions of interest (ROI) of camera views at two contrasting sites separated by 10 ° latitude (Figure 3).

Mean green fraction of regions of interests (vegetation) in images has been found to vary by illumination conditions (Ahrends et al., 2009). It has been reported that the systematic changes of image based colour indices during the season can be largely omitted by focusing the analyses on midday images (Ahrends et al., 2009), possibly by aggregating information from nearby days (Sonnentag et al., 2012), and by selecting a robust estimator that is aggregated, e.g. median. Colour



changes in deciduous vegetation and snow cover over the season are distinctive enough to be clearly seen behind the day-to-day variation due to clouds and irradiation (Peltoniemi et al., submitted, Arslan et al., submitted), but in wetlands and coniferous trees these differences can be more obscure.

Green chromatic coordinate, GCC = G/(R+G+B), was used as the colour ratio. Mean GCC of pixels in the region of interest (ROI) of an image were calculated. Individual pixels were accepted for the mean calculation if exposure was sufficient, i.e. pixels were not dim or overexposed (Sonnentag et al., 2012). We required that pixel digital numbers for R, G, and B were all in the range [30, 254]. Daily images were classified to classes 1, 2, and 3, based on sun elevation angles falling into categories > 30 ° [20°, 30°), (0°, 20°), respectively. Daily medians of GCC in these groups were calculated. In addition to these sun elevation class medians, we also calculated daily median from midday images between 10:00 – 14:00 UTC+2. Daily time series were subsequently smoothed with loess regression so as to present interpolated daily estimates because the number of the images for median calculation was small.

As Kaamanen is a northern site with no daylight during the winter time, there is a long gap in image data (Figure 4). The overall trend in the phenological development of the birch (*B. pubescens*) and other wetland vegetation (mosses, sedges, shrubs) is very similar among the sun elevation categories. The differences between these classes are clearly smaller than any changes related to phenological greening up of mosses and other wetland vegetation and even more limited than the increase of GCC due to budburst of dwarf birches (Mid May-Early July). The effect of snowmelt is seen as a quick decline of GCC in Birch and Wetland GCC in early May of 2015 and 2016. A reference panel shows variable response during the time period of November to April. The panel was not snow covered in April (inspected visually from images), which implies the variable response of the panel's colour index may reflect the profound change of the light environment due to snow melt of the surroundings and multiple reflections.

In Parkano, located in southern Finland, there are gaps in the data during the winter only in the classes 2 and 3 (Figure 5) because of higher sun elevations during the winter. Again all categories of sun elevation give similar overall development of GCC, including midday median. The greenness change from winter to summer is nearly as much for the planted non-native Serbian spruce (*P. omorika*) as it is for birch (*B. pendula*). However, GCC estimated for spruce by sun elevation class shows across class differences and lower GCC for higher sun elevation angles, which could be related to the darker surface of spruces foliage and how colour ratios are reproduced at low digital numbers in the shaded parts of the canopy.

## 5 Data availability

To distribute image time series, we established a community (*Phenological time lapse images and data from Monimet EU Life+ project (LIFE12 ENV/FI/000409),* https://www.zenodo.org/communities/phenology_camera/) in Zenodo repository

service (https://www.zenodo.org/) maintained by CERN. The service was established in the EU OpenAIRE project (https://www.openaire.eu/). The repository is meant for permanent archiving and distribution of research materials.

The image time series from each camera covers period between 2014 and 2016 or parts of it (Figure 6). All images are unprocessed. Time series are in their original quality, and only major status issues influencing analyses are reported in the

associated camera datasheet (https://doi.org/10.5281/zenodo.777952). Users are welcomed to report new status issues with the image time series for their contact persons (see camera datasheet). Time series coverage of images varies by site and camera, depending on the mounting date, and data connection variability at the sites, which occasionally terminates half-hourly submissions of the images to the ftp server in remote locations when cameras are on mobile phone network.

Image data is organised in camera-specific collections visible in our Zenodo community, each having their unique DOI-

number. Collections include images from each camera form 2014–2016 packed into a zip-file, named after camera ID and years of coverage. DOI of image zips are listed in Table 1.

Camera-specific information is available in camera data sheet also stored in Zenodo. For the present submission of 2014 – 2016 images of cameras in the Table 1, see the camera datasheet (https://doi.org/10.5281/zenodo.777952). We plan to update

the collection of images in the future, by adding new versions of the image datasets. For newer images not yet uploaded to Zenodo, please contact camera contact persons in the data sheet, and meta-data of present camera image uploads.

**6 Conclusions**

Cameras have shown their value in various phenological applications. Time lapse image series are rich in features, which could be analysed for scientific purposes, particularly for those addressing changes in time. Here, we presented a camera

network operating in Finland, with cameras mounted in sites ranging from hemi-boreal to sub-arctic regions. So far, we have used the cameras in the analyses of latitudinal trends of birch phenology (Peltoniemi et al., in revision), snow cover (Arslan et al., 2017), and analyses of Scots pine and wetland $CO_2$ fluxes (Linkosalmi et al., 2016) with encouraging results. Several possibilities for further research exist, particularly associated with temporal changes of colours in conifer canopies, and changes of snow cover.


So far, mostly simple analyses based on colours have been used in phenological and snow analyses, and we find them useful in tracking seasonal phenomena, particularly as they have been shown to be robust to light conditions here and elsewhere (Ahrends et al., 2008). Some aspects of colour analysis require further research, including the role of light scattering from snow and associated light environment change, which seems to complicate analyses of colour ratio changes of vegetation

targets. Some of these challenges could be overcome by combining radiation transfer approaches to image analyses (Wingate et al., 2015).

It is also interesting to note that image analyses and tools are developing quickly. Efficient image processing libraries are available in many languages, e.g. in C++ (OpenCV, with bindings in Python), and also more high level languages such as R and Python, rendering them useful for any researcher. For example R libraries such as greenbrown and phenopix group

statistical methods useful for phenological analyses of images (Forkel and Wutzler, 2015; Filippa et al., 2016). Research on the automation of more complex analyses based on structural features, on the other hand, could allow more possibilities and more reliable quantification of phenological and snow events and that vary by position within the image view, possibly automating a large part of the analyses (Filippa et al., 2016).

Networked cameras can be used for obtaining broad spatial information about ecosystem status, and thus to support earth observation and carbon balance and hydrological analyses. Inevitably, the spatial coverage of cameras deployed at research sites remains limited, so other means of extending the spatial coverage should also be considered. Increasingly, the number of cameras plugged into the internet by citizens and other actors outside the research community (see. Morris et al., 2013) could be useful for these purposes. These cameras rarely follow restrictions or recommendations of phenological camera

networks, but could still provide useful material for global mapping of vegetation and snow. Networked cameras at research sites providing quality assured material on phenological progress could support analyses of even wider networks of cameras using approaches based on machine learning and artificial intelligence.

## 7  Author contribution

MP coordinated the design of the network. MP, MA, PK, KB, AA participated in the design of the network and this study.

Funding was acquired by AA (coord.), MP, MA, PK, KB, MP, MA PK, KB, JL, JN are responsible for site specific installations. AA and CT prepared web interface for image downloads. MP wrote the study and MA, PK, KB, AA, JPT and JL contributed to the text. ML, JK participated in installations of cameras with other authors.

## 8 Competing interests

The authors declare that they have no conflicts of interest.

## 9 Disclaimer

Authors do not perform quality control of images. Experiences of data are collected by individual studies and authors request users to report their experience through the Monimet web site's interface.



## 10 Acknowledgements

With the contribution of the LIFE+ financial instrument of the European Union (LIFE12 ENV/FI/000409 Monimet, http://monimet.fmi.fi). We thank Joanna Norkko and Antti Nevalainen (University of Helsinki) for help in installations in Tvärminne research station, Jussi Vuorenmaa (SYKE) and Martin Forsius (SYKE) for discussions about installations of cameras to FinLTSER sites and Timo Vesala (University of Helsinki) and Tuomas Laurila (FMI), and Päivi Merilä (Luke) for facilitating installations. We also thank Ari Ryynänen, Esko Oksa and Jouni Karhu for installations in Parkano, Tammela and Punkaharju and Paljakka sites, and Juuso Rainne (FMI), and Janne Levula (University of Helsinki) in Hyytiälä.

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

**Tables**

Table 1 Camera sites. Coordinates are in decimal degrees WGS84. Additional information of camera network and image time series from 2014–2016 can be found also using https://doi.org/10.5281/zenodo.777952

| No. | Site | Lat. Lon. | Altitude above | Mean annual air temperature | Ecosystem / Dominan | Camera ID / position / color space / DOI | DOI |
|-----|------|-----------|----------------|------------------------------|----------------------|-------------------------------------------|-----|





| | | | sea (m) | (°C) and precipitation (mm) | t species | | |
|---|---|---|---|---|---|---|---|
| 1 | Hyytiälä (2014-2016) | 61.85 24.30 | 180 | 3.5 / 711 | Scots pine | MC106 / Crown / RGB MC107 / Ground / RGB | 10.5281/zenodo.815559 10.5281/zenodo.815557 |
| 2 | Kaamanen (2015-2016) | 69.14 27.27 | 155 | -0.4 / 472 [d] | Wetland | MC128 / Ground / RGB | 10.5281/zenodo.815553 |
| 3 | Kenttärova (2015-2016) | 67.99 24.24 | 347 | -1.0 / 521 [d] | Norway spruce | MC114 / Canopy / RGB MC115 / Crown / RGB MC116 / Ground / RGB | 10.5281/zenodo.815519 10.5281/zenodo.815523 10.5281/zenodo.815521 |
| 4 | Lammi (2016) | 61.05 25.04 | 119 [h] | 4.3 / 644 | Canopy: mixed; Other cameras: Downy birch | MC122 / Canopy / RGB MC123 / Canopy / IR MC124 / Landscape / RGB MC125 / Landscape / IR MC126 / Ground / RGB MC127 / Ground / IR | 10.5281/zenodo.815538 10.5281/zenodo.815540 10.5281/zenodo.815542 10.5281/zenodo.815544 10.5281/zenodo.815546 10.5281/zenodo.815548 |
| 5 | Lompolo-jänkkä (2015-2016) | 69.80 24.21 | 274 | -1.0 / 521 [d] | Wetland | MC129 / Lompolojänkkä / RGB | 10.5281/zenodo.815555 |
| 6 | Paljakka (2015-2016) | 64.68 28.11 | 257 [h] | 1.6 / 918 [e] | Spruce stand | MC117 / Canopy / RGB MC118 / Canopy / IR [a]MC117-1/ Canopy / RGB | 10.5281/zenodo.815529 10.5281/zenodo.815527 10.5281/zenodo.815525 |
| 7 | Parkano (2015-2016) | 62.03 23.04 | 96 [h] | 3.5 / 667 [f] | Mixed | MC112 / Landscape / RGB | 10.5281/zenodo.815487 |





| 8 | Punkaharju (2014-2016) | 61.81 29.32 | 88[h] | 3.8 / 604 [d] | Norway spruce | [b]MC103/ Ground / RGB [b]MC104 / Crown / RGB [b]MC105 / Landscape / RGB | 10.5281/zenodo.815460 10.5281/zenodo.815462 10.5281/zenodo.815464 |
|---|---|---|---|---|---|---|---|
| 9 | Sodankylä, pine (2014-2016) | 67.36 26.64 | 179 | -0.4 / 527 [d] | Scots pine | MC108 / Canopy / RGB MC109 / Crown / RGB MC110 / Ground / RGB | 10.5281/zenodo.815479 10.5281/zenodo.815481 10.5281/zenodo.815483 |
| 10 | Sodankylä, wetland (2014-2016) | 67.37 | 180 | -0.4 / 527[d] | Wetland | MC111 / Ground Peatland / RGB | 10.5281/zenodo.815485 |
| 11 | Suonenjoki (2015-2016) | 62.64 27.05 | 135[h] | 3.9/ 566[g] | Scots pine | MC113 / Canopy / RGB | 10.5281/zenodo.815489 |
| 12 | Tammela (2014-2016) | 60.65 23.81 | 144[h] | 5.4 / 601[e] | Norway spruce | MC102 / Ground / RGB [c]MC101 / Crown / RGB MC100 / Canopy / RGB / | 10.5281/zenodo.815456 10.5281/zenodo.815454 10.5281/zenodo.815450 |
| 13 | Tvärminne (2016) | 59.84 23.25 | 3 | 6 / 634[d] | Mixed | MC130 / Landscape / RGB | 10.5281/zenodo.815550 |
| 14 | Värriö (2015-2016) | 67.75 29.61 | 400 | -0.5 / 601 | Scots pine | MC119 / Canopy / RGB MC120 / Crown / RGB MC121 / Ground / RGB | 10.5281/zenodo.815532 10.5281/zenodo.815534 10.5281/zenodo.815536 |

[a]Replaces MC117 11.2.2016 onwards.

[b]Re-positioned and refocussed Aug 2016 after damages in Jun 2016.

[c]Short period available from 2014 before camera damage.

[d]Long-term records of air temperature and precipitation for the period 1981-2010 for the station Hanko-Tvärminne (Pirinen et al. 2012).




[e]Weather station 'Puolanka Paljakka' data 2009-2017.

[f]Mean air temperature at weather station 'Somero Salkola', 2011-2017; precipitation sum 1981-2010 (http://www.metla.fi/metinfo/forest-condition/programmes/intensive-monitoring.htm).

[g]FMI Precipitation station 'Suonenjoki Iisvesi' data 2015-2016, temperature station measured at the site 1999-2010

[h]Altitude was determined from digital elevation model with a grid size of 10 m × 10 m provided by the National Land Survey of Finland.



**Figure captions**

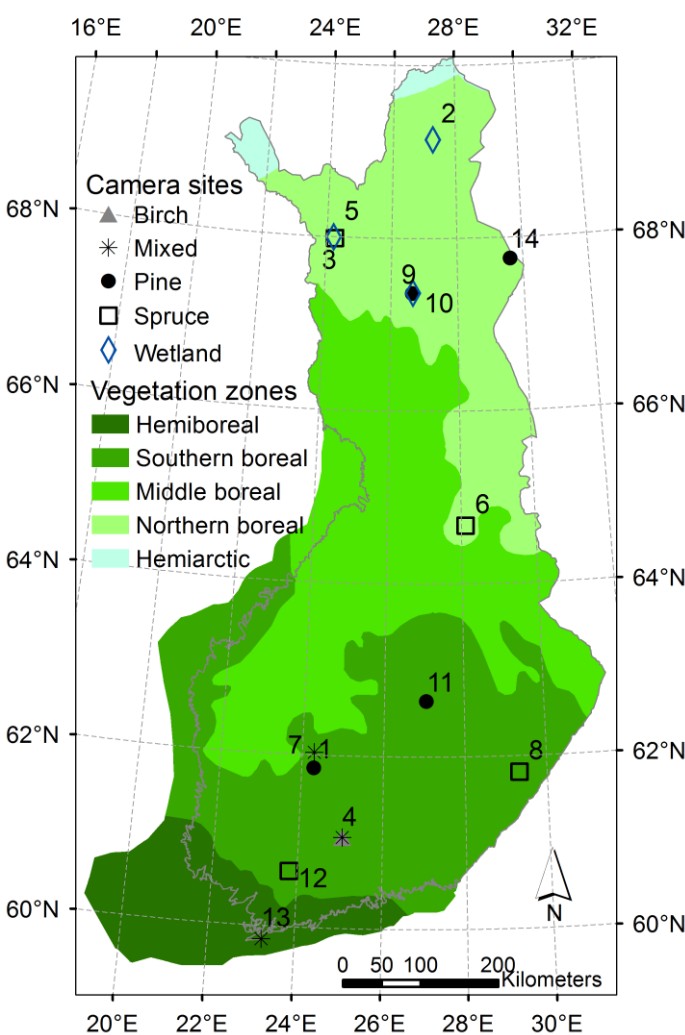

5    Figure 1 Sites in the phenological camera network, status end of 2016. Numbers refer to sites in Table 1. Data sources:
Country border © ESRI, Vegetation zones © SYKE, 2015.

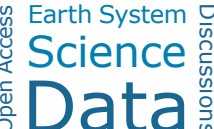




Figure 2 Camera views: MC100 (a), MC101 (b), MC102 (c), MC103 (d), MC104 (e), MC105 (f), MC106 (g), MC107 (h), MC108 (i), MC109 (j), MC110 (k), MC111 (l), MC112 (m), MC113 (n), MC114 (o), MC115 (p), MC116 (q), MC117 (r), MC118 (s), MC119 (t), MC120 (u), MC121 (v), MC122 (w), MC123 (x), MC124 (y), MC125 (z), MC126 (aa), MC127 (ab), MC128 (ac), MC129 (ad), MC130 (ae). Abbreviations refer to sites in Table 1.

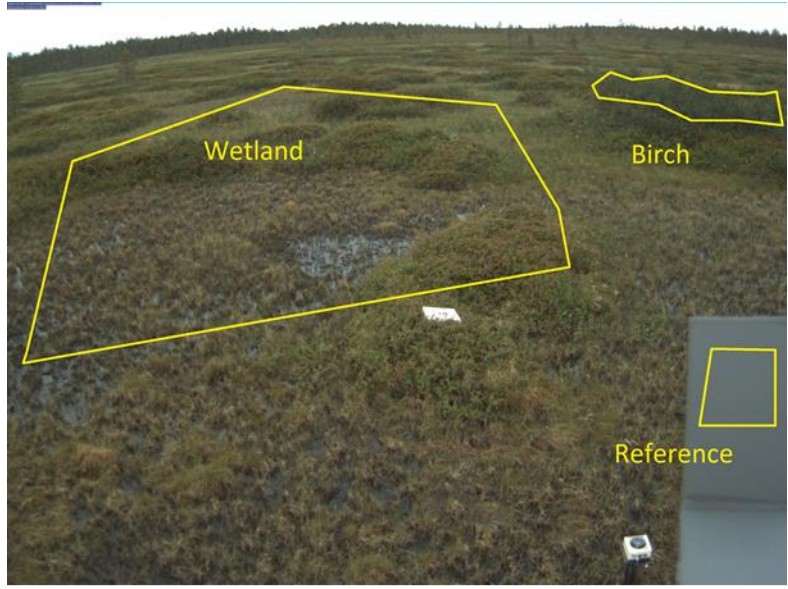

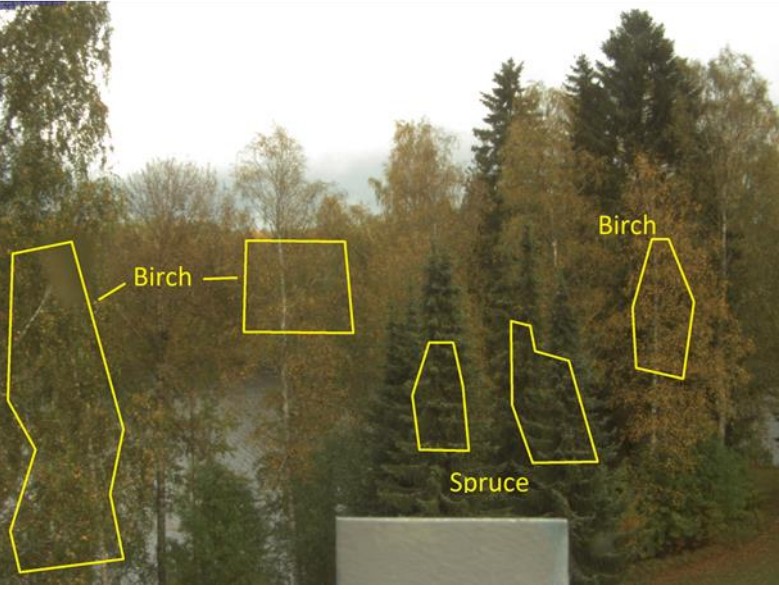

Figure 3 Regions of interest used in GCC time series construction. Top panel: Kaamanen wetland site, with Sphagnum spp and grasses (Wetland ROI), shrub like Pubescent birch (*Betula pubescens* subsp. Czerepanovii) (Birch), and reference gray



plate. Bottom panel: Parkano site, with Silver birches (*Betula pendula*) ROI distributed in three separated polygons (Birch), and Norway spruce (*Picea abies* L. Karst) ROI.



5   Figure 4 Daily GCC time series of ROI in Kaamanen site by solar elevation category, class 1: > 30 ° class 2: (20°, 30°), class 3: (0°, 20°), and as daytime averages (midday: 10:00-14:00), and smoothed development of GCC over the years.





Figure 5 Daily GCC time series of ROI in Parkano site by solar elevation category, class 1: > 30 °, class 2: (20°, 30°), class

3: (0, 20°), and as daytime averages (midday: 10:00-14:00), and smoothed development of GCC over the years.



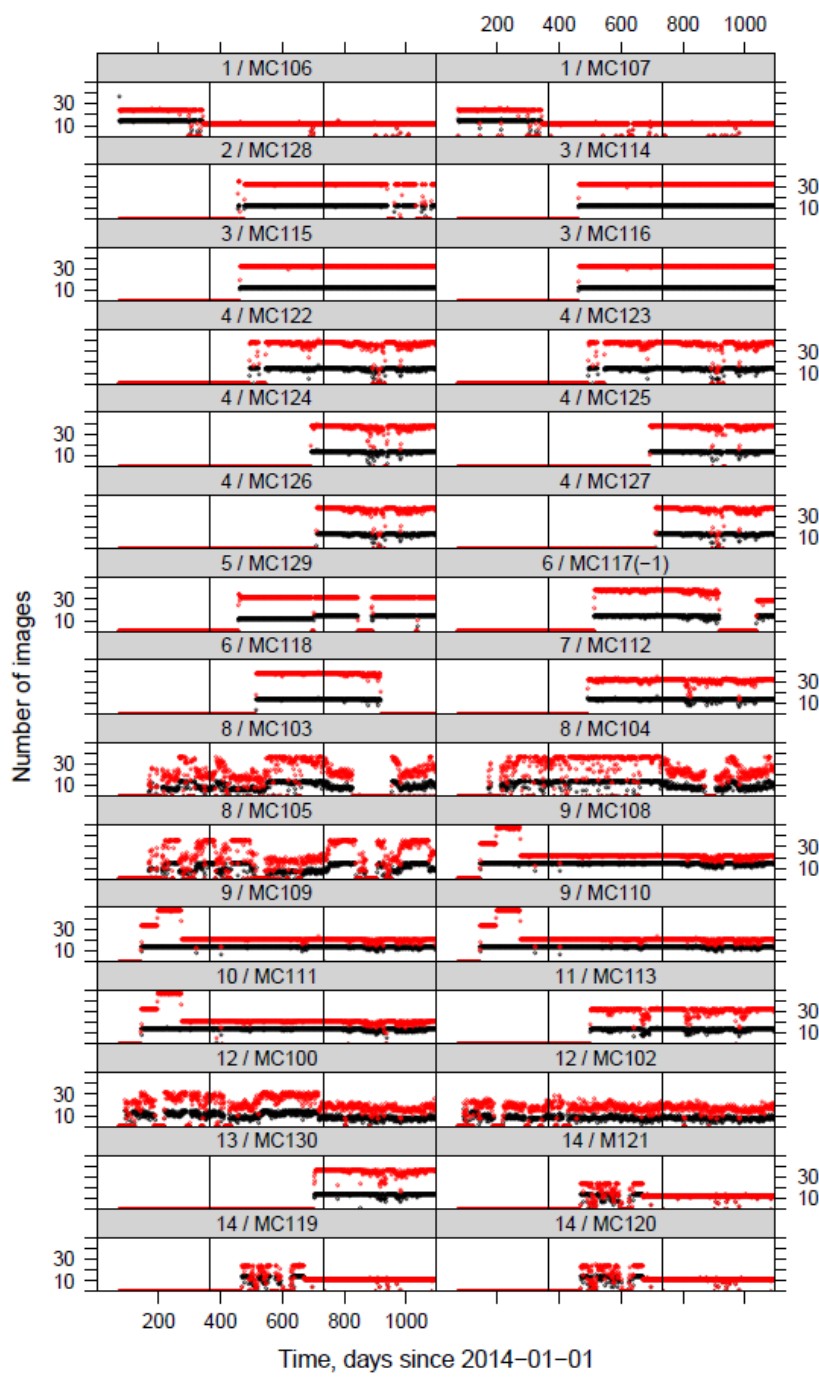

Figure 6 Availability of images. Images taken during the day do not cover darkest hours of the day. Number of images taken between 9:00–15:00 are in black, while all images taken during the day are in red symbols. Site labels refer to Table 1.