# Peer review of "Webcam network and image database for studies of phenological changes of vegetation and snow cover in Finland, image time series from 2014-2016"

_Earth System Science Data, 2017_

## Referee Comment (RC1) · Anonymous Referee #1 · 7 Sep 2017

The manuscript presents phenocam data from 14 sites belonging to the Finnish webcam network. Data collection methods and sites setup are exhaustively discussed. The manuscript provides all the information needed to other researchers for an easy and effective use of the data. Images are openly accessible and available from a to a long-term data repository. These data are relevant and image availability can be of great interest to the growing community of researchers using phenocams for phenological and ecological studies in northern ecosystems. Moreover also the hydrology community can benefit from these data for studies regarding snow dynamics.

[Figure]

I have few major comments:

1. In section 1, likely before the last paragraph I would add a couple of sentences mentioning shortcomings, difficulties and open issues of phenocam networks installation and maintenance such as sensor malfunctions, camera shifts, data transmission failure, hardware and software infrastructure needed , . . . .

2. The sun angle effect topic is interesting. It must be introduced in section 1. Moreover in section 4, I would mention that the results shown in fig4 and 5 (a negligible influence of sun angle with the exception of the spruce ROI in Parkano) confirm the correctness of using midday images for daily aggregation and filtering as proposed by several authors (Wingate et al 2015, Sonnentag et al 2012, Filippa et al 2016). In addition you should add that the sun angle effects you are presenting actually are the effects of the interplay between sun angle, canopy geometry and camera orientation. I'm not sure you would have observed a so important sun angle effect at the spruce ROI with a less vertical perspective of spruce trees.

3. In section 3.1, I would add a sentence on the filenaming rule used.

4. Lastly I must admit that many sentences are difficult to read, unclear and with many typo errors. I made a lot of small suggestions and corrections but I believe that English grammar, syntax and usage must be carefully revised and improved before acceptance.

I suggest manuscript publication after the above mentioned points and the following specific comments are addressed.

SPECIFIC COMMENTS

p1 l19-20 remove the first "and" after the comma and remove the last comma before the last "and"

p1 l24 I would remove the first link that is a specific link available at the data repository page www.zenodo.org/communities/phenology_camera
p1 l26 I'd suggest: ". . . image time series from cameras consists of . . ."

p1 l29-30 You are taking for granted that snow plays a role in regulating vegetation phenology that is not always true for forest species at least at global scale. You should give an ecological or geographical context to this sentence. e.g In northern ecosystems . . .

p2 l3 I'd suggest: "more monitoring data are required"

p2 l4 I understand the meaning but the sentence needs to be rephrased: what is "snow monitoring of phenology"?

p2 l6 I suggest adding "at high spatial and temporal resolution"

p2 l9 I'd suggest: ". . . has started to collect . . ."

p2 l10 some is plural: "some subregions are used to track color changes"

P2 l11 greenness indexes do not represent phenological changes of leaf color. They represent chromatic changes of the scene that are related not only to leaf color but also to leaf emergence/presence/abscission, morphology and conditions.

P2 l15 sentence unclear "and by selecting and colour indexing appropriately"? What do you mean? Selecting images or color indexes?

P2 l15-17 filtering explanation is unclear. Be more general simply citing the correct references or be more clear. "concentrating on well exposed midday images" means selecting those images? How is "good exposition" evaluated? Or it's simply averaging all images around midday. Moreover, at least Sonnentag et al 2012 method is based on GCC and not on RGB. So I would mention GCC first and than present the filtering approaches.

P2 l24 I'd rearrange this sentence a little: Budburst and leaf senescence of deciduous species and their relationship . . .

[Figure]

p2 l27 "the development of biochemical development" is awkward

p2 l31 add more references regarding the accuracy of phenophases estimation using phenocam.

p2 l32 Salvatori et al method is based on blue channel histogram thresholding.

P2 l34 Arslan et al, use the reference of the published paper

P2 l35 "time lapse material" ?? do you mean image archives? Quality assurance of what?. In this case you should mention that these objectives probably needs to be addressed by image visual inspection, as everything you mentioned before can be done automatically.

P3 l4 I'd remove ..."of cameras" + introducing/installing

p3 l15-16 ecosystem data is unclear. Maybe " . . . linking them to other remote sensing data or field observations'

p3 l21 "near-ecosystem remote sensing"? Do you mean "ecosystem proximal sensing"? Or ecosystem close range/near remote sensing ?

P3 l22 cameras monitoring ecosystems is awkward. Maybe ecosystem monitoring camera networks

p3 l26 I'd suggest rephrasing like "we further made publicly available image from 27 cameras of 14 sites in ...". I'd delete "established by EU (openaire and CERN)" as it is already mentioned at p10 l1-2.

P4 l3 I'd suggest rephrasing 'one to three cameras were installed at each site ..."

p4 l7 At this point of the paper the reader does not know the ecosystems investigated in the network. He is not aware of the existence of wetland sites. So I'd suggest to invert the order of paragraph 3.1 and 3.2 or at least move the first paragraph of section 3.2 at the begin of section 3.1

p4 l13 maximum, quarter to maximum resolution is unclear. You can provide image dimension (number of pixels).

P4 l17-20 I agree with you when you say that having different channel amplification settings with Stardot NetCam SC5, does not hamper inter-site analysis of greenness indexes temporal evolution. However, given the considerable effect of this channel amplification settings and considering US phenocam network recommendations (https://phenocam.sr.unh.edu/pdf/PhenoCam_Install_Instructions.pdf), I believe that a potential data user would be willing to know the settings of each camera. Are you providing these information somewhere?

P4 l17-18 the sentence is not correct. Channel amplification settings have not been modified by local conditions. I imagine that channel amplification settings have been modified from default values and adjusted depending on scene/site conditions. Moreover l18 what is a "typical" condition? Do you mean clear sky? Sunny?

P4 l22-23 "AXIS P1357E, similar settings were used, most importantly;" ? Check English grammar.

P5 l21 what is the aapa mire region? Is it the name of the region? Does it need to be written with a capital letter?

P5 section 3.2 Try to homogenize information given for each site. E.g. if you indicate LAI max or tree height data, you should provide them for each site.

P6 l21 I'd suggest rephrasing: "the camera was installed to provide"

p6 l25 Paljakka is a phenology monitoring ... . Add a

p6 l28/ p7 l3 the camera is located on the station roof

p7 l3 trees are observed/monitored not followed.

P7 l18 views of the forest canopy? .. on the forest canopy?

P8 l10 tree types. Maybe tree species is better.

P9 l1 to be clearly "depicted". Maybe better than "seen"

p9 l4-11 This paragraph is unclear, imprecise and must be rephrased: e.g. l6 pixels were not dim (?). how the range 30-254 was evaluated to omit under or overexposed pixels? I guess sun elevation angle was calculated using time of day and coordinates but you should at least mention it.

p9 l14-15 I'd suggest rephrasing: . . . the differences between sun elevation classes are negligible to changes caused by vegetation phenology . . .

p9 l16 I'd suggest to add a vertical line in fig 4 to indicate the date of snowmelt (even if visually estimated on the ROIs) in both years in all the panels.

P10 l4 time series or images? Normally time series refers to GCC time series and not to image stack. I would simply use images. Moreover what do you mean with unprocessed?

P10 l6 to their contact persons

p10 l9/l13 image data are organized / camera specific information are available . . .

p10 l18 maybe phenocam or digital cameras. Image series are rich in features is terrible. Maybe something along the line of "gather valuable information"

p14 table 1 header formatting must be verified and corrected (e.g DOI in column 7 and 8). a, b, c, d, . . . superscripts do not follow a logical order: d appears before a, b, c.

P19 fig2 You should consider to use camera id directly in the figure rather than inserting the correspondence in figure caption.

P20 fig3 insert camera id in the caption + shrubs like pubescent birch.

P23 fig6 I guess vertical black lines indicates different years? I do not see the need of showing red symbols as, if I'm correct, they include also night images that can not be

used for analysis.

---

## Author Comment (AC1) · 29 Sep 2017

Dear anonymous referee #1, We thank for your thorough review of the manuscript. We also thank for the positive comments, and are happy to implement your suggestions to the manuscript. Here we will shortly reply to the key points you raised, including the major comments and the selected specific comments that directly supplement the information content of the image material. We will implement fixes to the other specific comments in the next phase if the Editor allows us to do so. Major comment 1: We agree. There are practical issues, which the users of images, and also those planning

installing cameras need to know. We will phrase a short paragraph about it into the manuscript. Note also that we have collected known issues to the metadata-sheet of cameras that also has its own DOI. Major comment 2: Yes, we can introduce the sun angle analysis question in the introduction, and discuss previous methods in corresponding section of the manuscript. Major comment 3: Yes, we will add information about file naming rule. Major comment 4: We will do our best in improving the text (by accounting for the minor comments you had about the grammar), and ask native English speaker to improve the language. Among your specific comments, we selectively raised a few comments here: - P4 l17-20: we will add R, G, B amplification values to the metadata sheet of the cameras. - p4 l13: we will add pixel resolution of archived images to the metadata sheet of the cameras. - P5 section 3.2: We will review the consistency of data provided for the sites, and improve it. We may not be able to homogenize all information for all sites, as the sites have different data measured. We think it is still good to provide certain key data, e.g. LAI, for the sites we have it.

Best regards, Mikko Peltoniemi et al.
* * *

---

## Referee Comment (RC2) · Anonymous Referee #2 · 30 Oct 2017

The paper by Peltoniemi et al. presents an extensive network of automated cameras that are installed at field sites across Finland, spanning from the north to the south of the country. This paper is very useful for people who would wish to access this data for studies on the timing of phenological stages of vegetation. Furthermore, the paper includes a useful analysis of the daily variance in the GCC greenness index.

I have few comments other than those already raised by the other reviewer. My main point of criticism would be towards the claim in the abstract that this data can be used as 'ground truths to earth observations'. I agree that the timing of e.g. greening and

senescence can be derived from this camera network, which may be useful to interpret remote sensing data, but they can't act as a real ground truth. As is clear from Figure 2, many of the cameras are pointed horizontally towards tree canopies or aimed at the forest floor. Satellites, meanwhile, look straight down and see a completely different aspect of leaves, stems and the understory of the forest. This will lead to especially large differences when trees are partially covered in snow (as is the case at Hyytiälä, today on October 30th), but holds true for other parts of the year as well. Rather than suggesting that this data represents a ground truth to earth observations, I would suggest to say that this data can supplement earth observations.

Otherwise, I second the comment from the other reviewer that the English of this paper needs to be improved. The site descriptions are riddled with grammatical errors, but the rest of the paper could also benefit from a thorough language check.

Other, minor comments:
- Page 4, line 12: This sounds arbitrary. What determines whether an image is taken at full or a quarter resolution? And what is this resolution?
- Page 4, line 12: jpegs are compressed images by definition (that's how this format works). Do you mean the highest quality setting?
- Page 7, line 28: 'weather stations' should be singular, I assume?
- Page 9, lines 8-9: why choose medians? Was the daily pattern in GCC not normally distributed? How does it influence the data if a simple average is chosen?
- Page 9, line 21: I think you mean classes 1 and 2. Class 3 is the only one without a gap.

---

## Author Comment (AC2) · 6 Nov 2017

Dear Anonymous Referee #2,

We thank for your positive comments. Please find below our replies to referee crititisim.

Referee raised crititisism about our sentence stating that image time series could be used as 'ground truths to earth observations'.

We do agree that this is not the case, and we will reformulate this sentence. Cameras

indeed do not provide exact ground truths but rather timing information about phenomena, which can facilitate/supplement EO analyses.

Replies to minor comments:

Page 4, line 12: We will specify this can add information about camera resolution to the datasheet.

Page 4, line 12: We will also add information about quality setting to the datasheet of the cameras, and reformulate text.

Page 7, line 28: yes, will be corrected.

Page 9, line 8-9: Based on our experience use of average, median or some other percentile does no cause big difference in relative seasonal development of GCC. Here we chose median as it generally is the most robust estimator, given there may be occasional outliers in the observations.

Page 9 line 21: Yes, we do mean classes 1 and 2. We will correct this.

We will additionally proof read and correct grammatical mistakes in the manuscript with English native.

best regards, Mikko Peltoniemi et al.